# Late Pleistocene Expansion of Small Murid Rodents across the Palearctic in Relation to the Past Environmental Changes

**DOI:** 10.3390/genes12050642

**Published:** 2021-04-26

**Authors:** Katarzyna Kozyra, Tomasz M. Zając, Hermann Ansorge, Heliodor Wierzbicki, Magdalena Moska, Michal Stanko, Pavel Stopka

**Affiliations:** 1Department of Genetics, Wrocław University of Environmental and Life Sciences, Kożuchowska 7, 51-631 Wrocław, Poland; heliodor.wierzbicki@upwr.edu.pl (H.W.); magdalena.moska@upwr.edu.pl (M.M.); 2BIOCEV Group, Department of Zoology, Faculty of Science, Charles University, Viničná 7, 128 44 Prague, Czech Republic; pavel.stopka@natur.cuni.cz; 3Institute of Biological Sciences, Cardinal Stefan Wyszyński University, Wóycickiego 1/3, 01-938 Warsaw, Poland; tomzajc@protonmail.com; 4Senckenberg Museum of Natural History Görlitz, Am Museum 1, D-02826 Görlitz, Germany; hermann.ansorge@senckenberg.de; 5International Institute Zittau, Technical University Dresden, Markt 23, D-02763 Zittau, Germany; 6Institute of Parasitology and Institute of Zoology, Slovak Academy of Sciences, Hlinkova 3, 040 01 Košice, Slovakia; stankom@saske.sk

**Keywords:** *Apodemus agrarius*, environmental niche model, glacial expansion, Holocene bottleneck, MaxEnt, mitochondrial DNA, Muridae, phylogeny

## Abstract

We investigated the evolutionary history of the striped field mouse to identify factors that initiated its past demographic changes and to shed light on the causes of its current genetic structure and trans-Eurasian distribution. We sequenced mitochondrial cyt b from 184 individuals, obtained from 35 sites in central Europe and eastern Mongolia. We compared genetic analyses with previously published historical distribution models and data on environmental and climatic changes. The past demographic changes displayed similar population trends in the case of recently expanded clades C1 and C3, with the glacial (MIS 3–4) expansion and postglacial bottleneck preceding the recent expansion initiated in the late Holocene and were related to environmental changes during the upper Pleistocene and Holocene. The past demographic trends of the eastern Asian clade C3 were correlated with changes in sea level and the formation of new land bridges formed by the exposed sea shelf during the glaciations. These data were supported by reconstructed historical distribution models. The results of our genetic analyses, supported by the reconstruction of the historical spatial distributions of the distinct clades, confirm that over time the local populations mixed as a consequence of environmental and climatic changes resulting from cyclical glaciation and the interglacial period during the Pleistocene.

## 1. Introduction

Over the Pleistocene and Holocene, the global environment has undergone cyclical large-scale shifts that have affected ecosystems, species’ spatial structures and population trends [1,2,3,4]. The changing climate and global degradation of the environment resulting from human activities have both positively and negatively affected most global species. Deforestation, agricultural development, especially the cultivation of agricultural crops and the creation of a secondary (cultural) steppe, have positively affected the spread of several rodent species (European ground squirrel *Spermophilus citellus*, European hamster *Cricetus cricetus*, common vole *Microtus arvalis*, several mice species) from the Asian steppes to central Europe [5,6,7]. These pressures are unlikely to diminish in the future, so we need to acquire knowledge of the evolutionary ability of species to adapt to changing environments and their response to changes in the availability and extent of species preferred habitats, related to glacial and non-glacial cycles and more recently to human activity [8,9].

Whether or not a species can adapt to environmental changes strongly affects its vulnerability, thus influencing its genetic structure and shaping its distribution and spatial diversity [10,11]. For instance, in some invading populations of opportunistic or invasive species with a high adaptive capacity, such evolutionary changes can occur rapidly [12,13]. Reconstructing a species’ history and evolution in the context of concurrent environmental and climatic changes enhances our understanding of various aspects of its biology, such as how the species spreads and what factors affect this process, what role it plays in speciation processes, and its current population structure. Understanding the nature of a species’ successes and failures, especially when linked to particular events in its history, provides us with tools for making more reliable predictions of future changes, trends and threats in ecosystems affected by human factors and climatic changes [3,14,15]. Learning about the factors affecting the historical colonization of various species may provide insight into the processes behind current expansions, especially in the context of non-native and invasive species, to specify future trends, and to forecast the directions and scales of a species’ continued dispersion. 

Currently, we still observe progressive expansion of the striped field mouse (SFM) in Europe [14], the Siberian region [15] and far east Russia [16,17], closely related with human activity and habitat transformation leading to increased availability of the SFM’s preferred habitats and potential “new” migration corridors, allowing for overcoming the previously existing migration barriers. In the context of the observed high plasticity of SFM and their ability of fast occupation of newly formed preferred biotopes, this species should be in newly inhabited regions, where it appears as a new, alien element of local fauna, capable to changing the community of small mammals and displacing other “native” species [18,19]. Some European studies highlighted strong interspecies interactions between SFM and *A. sylvaticus* [18,20] and occupying the habitat niche of sympatric species such as *A. uralensis*, very quickly becoming a dominant element of the small mammal community [19].

Additionally, small mammals constitute a suitable model group for reconstructing the history of ecosystems and biota affected by glacial and interglacial cycles [21,22,23]. Western Palearctic wood mice of the genus *Apodemus* were used for assessing demographic and environmental factors that affected their populations during the Plio-Pleistocene [24], and these results were used as a proxy in analyses of the history of European forests. Similarly, phylogenetic studies on the Eurasian genus *Mus* enabled researchers to reconstruct the historical backdrop of shrinking forests and expanding grasslands [25,26]. 

The striped field mouse (*Apodemus agrarius*), hereafter SFM, is one of the most widespread rodents with an Eurasian distribution extending from eastern Asia to central Europe [27]. This distribution has an unusual pattern: its populations inhabit two disjunctive parts of Eurasia. This disjunct distribution may be a result of past demographic and spatial declines, as well as the fragmentation of the species’ former range following the original transcontinental expansion, and postglacial environmental changes, during the Postglacial period when the boreal forests, avoided by the species, replaced steppe and scrubland biomes preferred by SFM. This scenario of distribution disjunction, as a consequence of climate/environmental changes, is supported by the estimated time of disjunction dated at the early/middle Holocene (<12 ka) as the consequence of forest succession replacing previously dominant semi-open and open biota preferred by SFM or related with the climatic events during the MIS3, preceding the LGM (~39 ka) [28,29]. The former context is usually associated with Neolithic agricultural expansion [30]. The recently confirmed presence of the species in the region of Transbaikalia previously indicated as uninhabited or inhabited by a very scarce SFM population indicates current expansion in this region, mostly along the river valley and agricultural areas [15]. Some recent papers reconstructing SFM history in Europe using fossil records have suggested multiple probable SFM expansions, with a maximum frequency during the post-Neolithic period preceded by an early or middle Holocene population decline [6,7,31,32]. On the basis of their results of phylogenetic studies of SFM, Latinne et al. support the hypothesis of a relatively rapid expansion of SFM initiated in central Asia during the Eemian interglacial (MIS 5), when the open and semi-open habitats preferred by SFM were distributed across central Asia and Siberia, and further expansion into western Europe during the last Glacial period [29]. However, there is still no detailed analysis of the past SFM expansion and the factors that initiated it, especially in the context of its historical expansions of its Asian populations into the western Palearctic during the Pleistocene and Holocene.

Here, we analyzed the evolutionary history of the SFM and tested various scenarios of past events explaining its current genetic structure and current trans-Eurasian distribution in relation to past environmental changes during the upper Pleistocene and the Holocene. In doing so, we examined whether these climate-induced environmental changes could have influenced or initiated the large-scale trans-continental expansion of the species as well as past demographic changes within the different, site-specific populations, which in turn could have initiated colonization of new, previously uninhabited areas, and subsequently, spatial expansion and/or decline of populations as an effect of the appearance or disappearance of appropriate environmental conditions in relation to past climatic shifts.

## 2. Materials and Methods

### 2.1. Sampling

This study used newly collected genetic material from 361 trapped mice and museum specimens obtained from 35 sites in central Europe (Czech Republic, Slovakia, Germany, Poland) and Mongolia. The material included hair roots and tissue samples taken from museum specimens and root hair samples were collected from SFMs captured in Poland in wooden live capture traps between 2014 and 2015. The captured animals were marked and then released back into the wild. The sampled hair roots were stored in a freezer at −20 °C, and the tissue samples were stored in 96% ethanol until DNA extraction using a Sherlock AX DNA isolation kit (A & A Biotechnology, Poland). The procedure for catching and collecting the animals was approved by the Second Local Ethics Committee for Animals, Wrocław University of Environmental and Life Sciences (Nr 40/2012, 16 April 2012).

### 2.2. Genotyping

We analyzed the mitochondrial DNA variability of SFM using cytochrome b (cyt b) sequences from Eurasia, including newly sequenced specimens from Europe and Mongolia and other sequences from GenBank (Appendix A). 

We used two universal primers L14724 and H15915 for genotyping cyt b in the genus *Apodemus*, in each PCR reaction [33]. Each PCR contained 6.5 μL of 10X DreamTaq buffer with MgCl2 at a concentration 20 mM, 2.5 μL of dNTP mix (0.2 mM of each dNTP), 1.3 μL of forward and reverse primer (10 μM/1 μL), 0.23 μl of DreamTaq DNA polymerase (Thermo Scientific), 1 μL of genomic DNA, and 10.2 μL of ddH2O. The PCR comprised the following steps: 94 °C for 3 min, 35 cycles of 94 °C for 1 min., 55–57 °C (incremental increase in each cycle up to 57 °C) for 1 min., 72 °C for 1 min., and 72 °C for 5 min. PCR products were Sanger sequenced in forward and reverse directions using BigDye™ Terminator ver. 3.1 technology (Life Technologies, Carlsbad, CA, USA) on an ABI Genetic Analyser 3130 (Life Technologies). The sequences were manually edited using Bioedit to generate consensus DNA sequence from forward and reverse sequence [34] and aligned using the Clustal W method [31], implemented in MEGA 7 [32]. 

### 2.3. Phylogeny, Population Structure and Genetic Variability Based on mtDNA

We generated phylogenetic trees using Bayesian inference in BEAST ver. 1.8.4 [35], employing the birth–death process as a speciation, with *Rattus* and *R. norvegicus* (GenBank accession nos. AB752989 and KC735129, respectively) as an outgroup. To select the best-fit model of nucleotide substitution, we used the Bayesian Information Criterion (BIC) in MEGA 7, with the one partition consisting of three codons (positions 1–3) of cyt b [32]. The TN93+I+G model had the lowest BIC and AICc values; hence, it was selected as the best evolutionary substitution model. The MCMC chain with 100 million generations and tree sampling every 10,000 generations was used in three independent runs. The resulting log files were examined for convergence and effective sample sizes (with ESS > 200) in Tracer 1.7, and a maximum clade credibility tree was calculated in TreeAnnotator v1.8.4. A combined tree file was generated in LogCombiner v1.8.4 using a 25% burn-in period. Minimum spanning networks (MSN) for cytb sequences were prepared in PopArt ver. 1.7 [36]. Several of the shortest sequences were excluded in network reconstruction. 

For all the regions (continental populations) and clades designated by phylogenetic analysis, we calculated several parameters of genetic diversity using DnaSP ver. 5 [37] and Arlequin ver. 3.5 [38,39] (Table 1). Analysis of molecular variance (AMOVA) was used in the assessment of the geographical pattern of population subdivision [39]. The Genetic Landscape Shape Interpolation (GLSI) tool implemented in Alleles In Space (AIS, ver. 1.0), which allows visualization of patterns of genetic diversity across the landscape, graphically represent the spatial shape of the species’ diversity [40]. Additionally, we employed allelic aggregation index analysis (AAIA) to test whether the alleles showed non-random spatial diversity across the landscape studied. To test the significance of R_AVE_ (the mean AAI value), 1000 permutations were used. Correlations among geographical and genetic distances (IBD) were estimated with the Mantel test, using AIS. 

### 2.4. Historical Demography of Populations

We used three different approaches to analyze the demographic history of the SFM based on cyt b sequences. The first uses the mismatch distribution (MMD) of pairwise differences [41,42], analyzed using the pure demographic expansion and spatial expansion models fitted in Arlequin ver. 3.5 [38,39]. Both models assume the growth of a stationary population size from N0 to N1 during T generations. The pure demographic expansion model assumes the growth of a panmictic (global) population, whereas the spatial expansion model uses the infinite-island model (equivalent to the continental-island model), involving spatial range expansion and spatially diverged populations [38]. For both the sudden and spatial expansion models, we determined whether the sequences deviated significantly from the population, using the sum of squared deviations (SSD) and raggedness index (r) [41], estimated in Arlequin.

We also tested the demographic history using three neutrality tests: Tajima’s D test [43], Fu’s Fs statistic [44], determined using Arlequin ver. 3.5, and the Ramos-Onsins and Rozas R_2_ statistic [45], determined using DNAsp ver. 5 [37]. The statistical power of these tests depends strongly on sample size (n) and the number of segregating sites (S). The statistical significance of all the tests was estimated using 1000 permutations. The time of the last expansion was calculated on the basis of the Tau (τ) value generated in Arlequin using the formula t = τ/2 u, where t is the time (in generations) since the expansion and u is the cumulative evolutionary rate per generation for the sequence analyzed [46]. The evolutionary rate u is defined by formula u = 2 µk, where µ is the evolutionary rate (substitutions per site per generation) and k is the sequence length [42]. To calculate the expansion time in years, t is multiplied by the generation time of the species studied. According to a recent recommendation, we used the length of generation time g = 1.38 [47]. After taking into account time-dependent evolutionary rates, the time of the most recent expansion was estimated using the standard evolutionary rate of 0.024 substitutions per site as per My [48], and the recommended faster molecular clock of 0.027 to 0.036 substitutions per site as per My, as recommended for *Apodemus* populations with divergences of 130 ka or older [49,50].

The second method of demographic analysis we used was the Bayesian skyline plot method (BSP), implemented in BEAST software ver. 1.8.4 [35,51]. A powerful technique for inferring historical demographic changes based on sequence data [52], BSP helped us to visualize demographic changes in SFM populations over time. To identify phylogenetic clades, the analysis used the HKY+G+I model with a chain length of 150 million generations and a linear growth rate [51]. All of the parameters analyzed (the mean values with 95% confidence intervals) were estimated using Tracer software ver. 1.7; this was also used to check the results of the posterior distribution and to confirm that the effective sample sizes (ESS) were larger than 200 [53]. The Bayesian relaxed-molecular clock method was used to estimate divergence times while accounting for changes in evolutionary rates through time. A log-normal distribution was specified as the prior mean mutation rate, with a mean of 0.02, log (SD) of 0.56, and offset of 0.0172; it covered the range from 2% to 8% substitution per site per million years, with the 95% HDP (highest posterior density) range of 2.4% to 6.01% per Ma [54,55].

To corroborate the results with the past expansions reconstructed by BSP analysis, we tested several distinct demographic scenarios that could have occurred during the last glaciation and postglacial periods using an approximate Bayesian computation (ABC) framework in DIYABC ver. 2.1.0 [56]. The ABC approach allows a quantitative evaluation of the demographic and evolutionary history by strictly contrasting realistic models defined a priori and estimating relevant parameters [57]. The scenarios in both models included the constant population, i.e., null hypothesis, glacial expansion or decline, recent expansion, Holocene bottleneck followed by expansion, and middle/late glacial expansion followed by decline (Appendix A). The prior distribution of demographic parameters are listed in Appendix A. For each scenario within the two models, we calculated summary statistics that were used for comparing the simulated and observed datasets. Each model was run through 0.5 million simulations for each scenario, with summary statistics and principal component analysis used to confirm the good fit of all the scenarios with the observed data. These competing scenarios were then compared by estimating their posterior probabilities using polychotomous logistic regression on the 1% of simulated datasets closest to the observed data [58]. The best scenario was chosen based on the highest posterior probability with a 95% confidence interval (CI) not overlapping with the posterior probabilities of other scenarios; the posterior predictive error was calculated to evaluate the confidence of each scenario. The posterior probabilities of each parameter under the chosen scenario were analyzed using a local linear regression on the 1% closest simulated datasets, and logit transformation was applied to the parameter values [58]. An additional historical model (Model 3) with eight potential scenarios was prepared to reconstruct the possible events affecting the evolution and divergence of SFM in SE Asia in the context of its colonization of the Korean peninsula (Appendix A). The method of analysis in this model was as above.

### 2.5. Species Distribution Modelling

To estimate the historical distribution range of the SFM during the Last Glacial Maximum, we employed environmental niche modelling (ENM), using the maximum entropy method implemented in MaxEnt software ver. 3.4 [59]. MaxEnt reconstructs the distribution of a species based on presence data used as georeferenced point occurrences, and on annexed environmental condition data used as variables determining the species’ occurrence. Environmental data regarding actual and historical bioclimatic conditions were retrieved from the World Bioclim database, ver. 1.4. (http://www.worldclim.org, date of access: 29 April 2019), with a resolution of 2.5′ [60]. To avoid problems with collinearity among 21 bioclimatic variables, we eliminated highly correlated variables using the SDMtoolbox in ArcGIS (Appendix A). The final set of variables to be used in the niche modelling included twelve bioclimatic and three topographic variables (Appendix A). A detailed description of the method used for modelling the predictive past distribution of SFM is given in Appendix A and the measures of model accuracy in Appendix A.

The estimated areas of the species’ distribution during the LGM can be interpreted, especially in the temperate climate zone, as the refugia that members of the species would most likely have used under the most unfavorable climatic conditions. All the analyses were performed to reconstruct the historical distributions of SFMs across the whole of Eurasia as well as the identified clades (potentially capable of showing distinct habitat/climate preferences) with the confirmed strong signal of a recent expansion during the last glaciation modelled separately.

## 3. Results

### 3.1. Phylogenetic Analysis Based on Mitochondrial DNA

This analysis resulted in 184 newly sequenced specimens: 153 from central Europe (the Czech Republic, Germany, Poland and Slovakia), representing 59 haplotypes, and 31 specimens from eastern Mongolia, representing 19 haplotypes. Additionally, 134 haplotypes (289 sequences) were obtained from GenBank (Appendix A). All of the new haplotypes were submitted to GenBank (Accession Number MT113485-MT113569). Phylogenetic analysis indicated that the populations from Taiwan and Jeju Island clearly differed, the former consisting of the monophyletic clade C7 and the latter of clade C6. Together, the Eurasian continental populations (not including the two island populations) were classified into five clades: four strictly Asian (C2–C5) and one distributed across the Eurasian northern hemisphere (C1) (Figure 1 and Figure 2). Even so, these classifications did not reflect the populations’ biogeographic regions (Figure 1), with specimens from China and Mongolia being present in all of the Asian clades (C2–C5) (Appendix A). Specimens from South Korea are represented mostly in C3. The distributions of the specimens classified into clades C4 and C5 overlap and cover mostly central and eastern China, with the northernmost locations being near Primorski Krai in the Russian far east and the southernmost ones below 30° N. The easternmost range limit of clade C5 includes the remote islands along the west coast of South Korea. Clade C3 encompassed all the specimens from South Korea and some from remote islands, but its range extends to eastern and north-eastern China, western Mongolia, and the far east of Russia. Haplotypes from clade C2 are distributed over a large area of central and north-eastern China to the north of the Russian far east (Khabarovsk Krai and Magadan Oblast). The northern hemisphere clade C1 included specimens from Europe, the Siberian region, and north-eastern Asia above 40° N (north-eastern China, eastern Mongolia and the far east of Russia) (Figure 1).

### 3.2. Population Structure and Genetic Variability

All of the SFM specimens had a mean haplotype diversity (Hd) of 0.98 and nucleotide diversity (π) of 0.013. The Asian populations had the highest nucleotide diversity (π = 0.0156) and the average number of nucleotide differences (k = 17.804), with particularly small values for China (π = 0.011 and k = 12.849) and Mongolia (π = 0.0115 and k = 13.097). The Asian SFM specimens from China and Mongolia exhibited a higher genetic diversity than those from western Europe (π = 0.004, *p* < 0.05) and from the far east Russia (Appendix A). The Korean and Russian populations had much lower π and k values, similar to those observed in the populations from Taiwan and Jeju Island (Appendix A). European populations had the lowest π and k values (π = 0.004 and k = 4.523), with π ranging from 0.0013 to 0.007 and k from 1.46 to 6.44 (Table 1). The Eurasian clade C1 was lower in genetic diversity indices than Asian clades C3–C7.

The Mantel test revealed a pattern of isolation by distance (IBD), i.e., positive relationships between pairwise genetic and geographical distances, for the entire region studied (r = 0.31, *p* < 0.001). Clades C1 and C3 had higher IBD signals (r = 0.34 and 0.49, respectively, both significant at *p* < 0.001), as well as Eurasia; after excluding the island populations from Jeju Island and Taiwan (r = 0.39, *p* < 0.001), however, Europe did not exhibit the IBD pattern (r = 0.08, *p* = 0.69), though Asia did (r = 0.35, *p* < 0.001). The AAIA indicated significant allelic aggregation in the Eurasian population (R_AVE_ = 5.608, *p* < 0.001) and in clades C1 (R_AVE_ = 14.287, *p* < 0.001) and C3 (R_AVE_ = 3.702, *p* < 0.001), suggesting a non-random distribution of genotypes. The GLSI revealed a cline of genetic diversity of SFM from eastern Asia to Europe, with the highest diversity in Asia, and decreasing diversity towards the east and north-east in clade C3 (Figure 3).

AMOVA testing on the seven genetic groups (C1–C7) showed that the differences between the clades accounted for 68% of the total variance (Table 2), with the percentage genetic variation within and between populations expressed in terms of overall genetic variance. The molecular variance between the clades in the continental population that were divided into three groups (Eurasia, Taiwan and Jeju Island) was higher than the intra-population (clade) variability (39.8% versus 25.1%). Hierarchical AMOVA conducted for the 11 populations defined by countries (with the Jeju Island population treated as a distinct group) and classified within three isolated groups (one continental Eurasian and two island populations, Jeju Island and Taiwan) indicated a 50.1% genetic variation between the groups (versus 30% of total genetic diversity for the clades), with only an 18.8% variation between the nine populations from continental Eurasia (Table 2). These results suggest that genetic diversity within country populations was higher than the diversity between countries, and similarly, that molecular variation within the clades was greater than the variation between the clades. These results suggest a strong genetic structure between the clades.

### 3.3. Demographic Analyses

Both the spatial and sudden demographic expansion models, as well as the neutrality tests, displayed the population growth of the SFM, as indicated by the strong signal of a recent SFM expansion in Eurasia, Asia and Europe (Appendix A). This signal strength was determined only for C1 and C3, the two most recently diverged clades. In both the spatial and sudden demographic models for Eurasia and clades C1 and C3, SSD and Harpending’s raggedness index r were statistically insignificant, the models showing a good fit between the observed and expected MMD values. Only for the Asian and European populations did the tests support the corresponding spatial expansion models. For clade C6, the models were well fitted, as indicated by the neutrality test showing low and non-significant values, different expected and observed MMD values, and goodness-of-fit tests (Appendix A). 

The sudden and spatial models estimated similar expansion times of the SFM in Eurasia (τ = 14.1 and 11.0, respectively). The models also estimated similar distributions of pairwise differences (Table 3, Figure 4). The Asian (mainland) and European population expansion times were quite different, as estimated by the corresponding spatial expansion models (τ = 15.5 and 2.1, respectively) (Table 3). The results of the historical demography in clade C1 indicated the estimated time of the most recent sudden expansion at ca 99–66 ka (MIS5), and the estimated time of spatial expansion at ca 64–42 ka (MIS4) (Table 3). Similarly, the estimated time of spatial expansion for clade C3 was slightly more recent than the estimated demographic expansion for this clade, with the most recent estimate (3.6% per Ma) dated to the early Weichselian (MIS 5a–d). The glacial expansion times of both clades were confirmed in the BSP demographic history (Figure 5), with a more recent rapid growth of clade C1′s population size during the MIS 4-3, and an earlier growth than clade C3 after the last Interglacial period during the MIS 5a–d (Figure 5). The population growth rate decreased, reaching negative values during the LGM and postglacial periods, as did the effective population size in all the populations analyzed. These rates, however, rapidly increased during the middle (Northgrippian) and late Holocene (Meghalayan) (Figure 5).

ABC analysis corroborated the scenarios of the middle Weichselian (MIS 4-3) expansion initiated during the early Weichselian (MIS 5) or earlier, and the postglacial population decline (Figure 6a, Appendix A). The best scenario in the first model (Figure 6a) recovered the ancestral population of clade C3 expanding (t_4_) at 97,300 (95% CI 51,600–119,000) generations ago (i.e., about 70–164 ka, median 134 ka) and at 63,500 (95% CI 46,000–114,000) generations ago (i.e., 63–155 ka, median 87 ka) for clade C1, with estimated population declines (t_2_) during and after the late Glacial at 20,500 (95% CI 9650–29,500) generations ago (about 13–41 ka) and 13,400 (95% CI 9200–28,000) generations ago (about 12.5–38.5 ka, median 18.5ka), for clades C3 and C1, respectively (Appendix A).

The best scenario in the second model (Figure 6a), with recent (late Holocene) expansion preceded by a postglacial bottleneck after the glacial expansion, was the one (Figure 6a) with late glacial/postglacial population declines (t2) at 15,900 (95% CI 9300–28,700) generations ago (about 13–39 ka, median 22 ka) for C3 and 15,000 (95% CI 9200–28,500) generations ago (about 12–38 ka, median 21 ka) for C1, and estimated early Weichselian (MIS 5a–d) or earlier expansion (t4) at 104,000 (95% CI 57,000–119,000) generations ago (about 78–164 ka, median 143 ka) and 87 600 (95% CI 50,200–119,000) generations ago (about 69–160 ka, median 120 ka) for C3 and C1, respectively. The Holocene expansion, after postglacial population decline (t1), is estimated at 5460 (95% CI 1540–11,600) generations ago (i.e., 2.1–11.6 ka, median 7.5 ka) and at 3360 (95% CI 1090–11,000) generations ago (i.e., about 1.5–15.1 ka, median 4.6 ka) for the clades C3 and C1, respectively (Appendix A). The scenarios with the constant population and glacial bottleneck were rejected in both models (Appendix A, Appendix A). 

An analysis confirmed the first expansion of SFM during the middle and late Weichselian (MIS 4–2) and postglacial population decline preceded the second expansion during the middle/late Holocene (~5–7 ka).

The reconstruction of SFM history in east Asia in the context of its colonization of the Korean peninsula (Model 3) included the specimens from China and Korean peninsula (clades C3 and C5) supporting scenario 4 with ancestral population represented by specimens from Korean remote islands and the Korean peninsula from the mainland China population diverged (t_2_) 218,000 generations ago (~300 ka), preceding the split of ancestral (initial) Korean population (t_1_) ~189,000 (95% CI 61,300–368,000) generations ago (~260 ka, 84.5–508 ka) (Figure 2, Appendix A) distributed along the (mainland) Korean peninsula. The other analyzed scenarios were rejected as less supported (Appendix A and Appendix A).

### 3.4. Species Distribution Modelling

The fitted model of SFM distribution, with two distinct clades (C1 and C3) highlighted during the LGM, indicated that these two populations had partially overlapped environmental niches (Figure 6). The reconstructed possible glacial refugia for clade C3 were limited mostly to south-eastern Asia, with the highest probability of occurrence on the exposed sea shelf between eastern China and the Korean peninsula, a result of the glacial fall in sea level (Figure 7; Appendix A). Similarly, the glacial ENM for clades C2 and C5 covered the Yellow Sea, the South China Sea and adjacent regions (Appendix A). However, the region with the currently highest probability of the presence of clade C5 covers much of south-eastern China, as well as the east coast of the Korean peninsula and the north-eastern China (Manchurian) Plain. In Europe, possible glacial refugia of clade C1 have been identified in southern and south-eastern Europe (Figure 7a). During the LGM, the Black Sea and Caspian Sea regions were the locations most likely to be inhabited by the northern SFM population, as well as the Balkan region and the Po valley in the north of the Italian peninsula, identified as the main glacial refugia of the SFM in Europe. The reconstructed glacial refugia for clade C3 in eastern Asia were mostly restricted to north-eastern China and the adjacent areas of the exposed sea shelf (Figure 7b). The results of the current and past distribution modelling of clades C1–C5 are presented in Appendix A.

Analysis of climatic factors (variables) used in the reconstruction of current and past ENMs indicate the very high impact of temperature-related variables on prediction in the case of clade C1 (with >60% contribution), compared to other, strictly Asiatic clades (C3–C5) for which the precipitation-related variables had an most important impact in model building (with >50% contribution) (Appendix A).

## 4. Discussion

The phylogenetic structure of Asian populations, with monophyletic clades representing the island populations from Taiwan and Jeju Island, supports prior analyses [73,74,75]. This genetic structure is a consequence of the islands’ geographical locations, which have undergone cyclical periods of isolation, together with a few relatively short periods when they were connected to the Asian mainland due to sea-level lowering in glacial periods. In the case of Taiwanese population (C7) two highly supported sublineages exist, distributed along the eastern and western part of island as a consequence of multiple invasions from continental Asia during subsequent glaciations and long-term isolation by mountain massif during the interglacial. Analogical multiple colonization is expected in the case of population from Jeju (C6). 

The genetic divergence between the European and Asian mainland populations was smaller than the divergence between the mainland and island populations of SFM, but equilibrium models of isolation by distance showed that genetic variance between sampling locations increased with their geographic distance. The westward significant decrease in genetic diversity observed in Europe supports the hypothesis of a relatively recent expansion into western Asia and Europe [76]. The farther the distance from the source site, the greater the between-group genetic diversity, but the smaller the within-group genetic diversity, consistent with the conclusions drawn from studies employing serial founder models of migration [77,78]. The SFM specimens displayed a similar geographic pattern of genetic diversity: an ancestral population from single location in eastern Asia that spread through eastern Russia and Siberia to Europe, with its genetic diversity decreasing with increasing distance from the area of origin, especially across the northern hemisphere. 

### 4.1. Demographic Analyses

Current knowledge of the demographic history of the SFM, especially in the context of its westward expansion into Europe through Siberia, is still insufficient. The demographic analyses of the whole sample demonstrated that the general SFM population showed a tendency to grow, as indicated by both the spatial and sudden demographic expansion models and confirmed by the neutrality tests. The indices of demographic history based on neutrality tests indicated signs of expansion for only two genetic clades, C1 and C3 (Appendix A). Nevertheless, the results obtained for clades C4, C5, and C7 could have been affected by their small sample sizes, making historical demographics difficult to analyze [79]. 

The two methods used in this study to estimate expansion time, mismatch analysis (MMA) and Bayesian skyline plot (BSP) gave different results of estimated expansion time for the Eurasian specimens of the SFM (Figure 5). There are two likely reasons for this discrepancy. First, the two methods use different techniques to reconstruct and interpret past demographic signals. Second, while deep coalescence does not affect BSP, it can affect MMA [79]; consequently, BSP estimated a later time of expansion than did MMA. We also observed particularly strong discrepancies in the results of the estimated changes in demography for the Eurasian and Asian specimens, which pooled individuals from different clades (Table 3). This discrepancy may have been due to either mixing of the differentiated specimens [79], incomplete or weak sorting of the clades [80], or both of these factors combined. The results of our genetic analyses, supported by the reconstruction of the historical spatial distributions of the distinct clades, confirmed that over time, the regional populations mixed as a consequence of environmental and climatic changes, affected by cyclical glaciation and the interglacial period during the Pleistocene. 

The demographic history of SFM, reconstructed using the BSP method, showed an early, long period of stability followed by rapid population growth during the last glaciation (Figure 5). This series of changes resulted in contemporary genetic sequences which had lost the genetic information from the pre-Glacial period [81,82]. It is important to remember, however, that when reconstructing historical demography and estimating the time of expansion, time-dependent evolutionary rates can strongly affect the results [52,83]. Many authors have discussed the methods of determining long- and short-term mutation rates [83,84,85]. Based on recommendations for *Apodemus* from previous publications, we chose the molecular clock calibration technique [49,50]. The results of Bayesian demography derived from using the BEAST software complied with these assumptions, with the estimated (median) molecular clock of 3.6 to 3.7% per million years. One should exercise caution when interpreting the results presented on the plot of demographic changes, however, especially the accuracy of the estimation of the time when population changes occurred. Such an interpretation should take into account historical environmental and climatic changes that were likely to influence the populations’ demography. Using the recommended faster molecular clock, both the MMD and BSM models yielded similar results for clades C1 and C3 (Figure 5). In both clades, the estimated spatial expansion time with a fast molecular clock fell within the period of growth rate in BSP (Figure 5). The Bayesian reconstruction of the historical demographic trends for the clades showed that there were two region-related periods of expansion within Eurasia, with the first one during the last glaciation and the second during the late Holocene.

### 4.2. The Demographic History of Asian Clade C3

The south-eastern Asian clade C3, probably distributed along the Korean peninsula and adjacent areas, diversified relatively recently. The first Korean fossil records, identified solely in the upper Pleistocene and Holocene deposits, indicate a recent expansion to the Korean peninsula [68] (Figure 6b). 

The reconstructed past demographic trends (BSP) of clade C3, with population growth during the lower (MIS 5a–d) and middle Weichselian (MIS 3–4), showed a strong correlation of the SFM’s history with sea level changes after the last interglacial (Figure 5c). This same scenario was corroborated by ABC analysis in the first and second models (Figure 6a, Appendix A). The GLSI model indicated that the species expanded from eastern China to the Korean peninsula and the far east of Russia, across the Yellow Sea and Bohai Sea regions (Figure 3c). This scenario is also supported by the reconstruction of the past (LGM) widespread distribution of SFM clades C5 and C3. These results indicate the important role that the exposed East China Sea shelf (ECSS) and Yellow/Bohai Sea shelf (YBSS) played during the glacial sea-level regression in shaping the species’ history and current genetic structure, a conclusion supported by the historical distribution models (Figure 7, Appendix A). During the glaciations, the exposed ECSS and BYSS formed a long-lasting land bridge between south-eastern Asia and the Korean peninsula [86]. This bridge became a migration corridor for the SFM, enabling it to colonize the Korean peninsula (Appendix A). The glacial large-scale expansion of the open and semi-open habitats and the retreat of forests in south-eastern China [87,88] created favorable conditions for animals to colonize regions previously deemed unsuitable [89]. The reconstructed historical vegetation of the exposed sea shelf during the last glaciation was dominated by open grasslands and freshwater wetlands [90,91,92]. The Older Dryas (a cold period) and the Younger Dryas (a dry period) were characterized by rapid changes in vegetation structure in the Yellow Sea region, from the dominance (during warm and humid interstadials) of woody C_4_ plants to the dominance of grassy C_3_ plants [93]. During the stadial (cold) periods, the vegetation on the north-eastern Chinese coastline consisted mainly of steppes and shrub-steppes dominated by *Artemisia* [94], with both biomes offering favorable conditions for the SFM to expand from south-eastern China. The Bohai and Yellow Sea shelf were still exposed during the warm Bølling/Allerød interstadial (~14.7–12.7 ka), characterized by the dominance of steppe biomes in the north China plain. Following transgression by the sea, the water level over the North Yellow Sea shelf quickly rose) at the end of the Younger Dryas (ca 11.5 ka) [95]. The scenario with exposed BYSS and ECSS was initially inhabited by migrants from eastern China, forming a new “source population” which diverged from clade C5 and then expanded into the Korean peninsula; this is corroborated by ABC analysis as the most probable scenario in model 3 (Figure 6b; Appendix A and Appendix A). The estimated split time of the ancestral population which spread from China and colonized the Korean peninsula (Figure 6b) is dated at MIS 8/MIS 9 (t_1_, ~300 ka), and preceded the divergence of the highly supported strictly Korean (peninsulan) population during the MIS 7/MIS 8—glacial/interglacial transition (t_2_, ~260 ka) (Appendix A. These results indicate an earlier divergence of phylogenetic lineage (clade C3) recently distributed along the Korean peninsula and remote islands, preceding the colonization of the Korean peninsula during the last Glacial period.

Results of recent and glacial ENMs indicate the presence of habitats potentially suitable for SFM in south Japan (Figure 7, Appendix A). Nevertheless, there is no evidence that the SFM has ever inhabited Japanese islands, even though they do offer potentially suitable habitats [96,97]. Recently, species inhabit only the Uotsouri Jima Islands, near Taiwan, connected with mainland China during the last glaciations [98]. This can be explained by a permanent barrier, formed by the Sea of Japan and the Korean (Tsushima) Strait, between Japan and the adjacent mainland regions, which made it impossible for the SFM to colonize these islands (Appendix A), even during the penultimate and the last glacial maxima [99]. Endemic Japanese murids—*A. argenteus* and *A. speciosus* colonized the Japanese islands no later than the middle Pleistocene, when the Tsushima land bridge was formed [100]. The Korean field mouse, *A. peninsulae*, colonized only the Hokkaido island during the upper Pleistocene glaciations, when the land bridge between Hokkaido and Sakhalin was formed. In both examples presented above, the colonization of the Japanese Archipelago can be placed before the SFM expansion into the Korean peninsula and far east Russia region. The presence of harvest mouse *Micromys minutus* in the Japanese Archipelago and more recent (~300 ka) colonization of southern islands (Kyushu, Shikoku, Honshu) is still discussed and requires an explanation [101]. The most probable explanation is dispersal by the Tsushima Strait on floating islands composed of plants [100], an unlikely scenario in the case of SFM.

### 4.3. Extensive Westward Expansion during the Last Glaciation

The recent westward expansion of SFM to central Europe has not been tracked in detail to date. A preliminary analysis based on the short genetic distance between specimens from Europe and the Russian far east permitted an approximation of the duration of the SFM’s westward expansion; according to Suzuki et al. [102], it was ca 200 ka, while Sakka et al. [76] placed it within the 175–190 ka range. Our analysis gave a similar estimated expansion time for the Eurasian samples. As previously mentioned, Suzuki et al. [102] identified the specimens from the far east of Russia and north-eastern China as an ancestral population that dispersed from eastern Siberia to central Europe (north of 40° N). Our reconstruction of the historical demographic trends of the northern hemisphere clade C1, however, suggest that they had expanded earlier than previously estimated, during the early Weichselian (MIS 5a–d). Standard and fast molecular clock calibrations estimated the time of this as between 100 ka (standard evolutionary rate) and 66 ka (fast evolutionary rate) for the sudden demographic expansion model, and between 64 ka and 42 ka for the spatial expansion model (Table 3). The Bayesian demographic population changes plot (BSP) supported the faster (3.6 × 10^−2^/Ma) evolutionary rate estimates by the MMD. 

To understand why the last glaciation allowed the transcontinental dispersion of the SFM to the west, we can analyze the species’ requirements in the context of environmental and climatic changes throughout its history. If we consider the range of glaciers during the penultimate (Saalian) glaciation (the most extensive one in Eurasia), we see that expansion during this period was unlikely (Appendix A). The glacial extension during the last Glacial period was much smaller than during the Saale glaciation (MIS 6), with a limited glacier range in Asia during the middle and late Weichselian (MIS 2–3) (Figure 6). 

The SFM inhabits mostly open and semi-open habitats (i.e., steppes, shrub-steppes, meadows, and arable fields) across northern Eurasia [14,15,102,103,104]. The reconstructed habitats in southern Siberia during the last Interglacial (~128–117.4 ka) were characterized by shrinking steppe plant communities and shrubby tundra, which were dominant at the end of the penultimate glaciation, and the expansion of boreal forests (mainly Scots and Siberian pine taiga) [105,106]. At the end of the interglacial period (~117.5 ka BP), forest areas across northern Eurasia were again replaced by steppe biomes and cool grass-shrub communities [105]. Palynological data and the reconstruction of LGM biomes confirmed that steppe and forest-steppe biomes in eastern Europe and southern Siberia during the last glaciation formed a migration corridor for the SFM [107] (Appendix A). At 15 ka, steppes and tundra remained the dominant Siberian biomes [108,109]. Environmental changes in the Siberian region during the Holocene, with open and semi-open biotas replaced by Siberian taiga, explain why its Eurasian range is separated into two disjunct parts by the Transbaikalia region: the expansion of the taiga.

The reconstructed vegetation of eastern Europe and Siberia during the middle and late Holocene showed that forest biomes at that time covered a noticeably larger area than they did during the glacial period [110,111]. The end of the glacial period (~14.5–12.65 ka BP) revealed a sharp increase in the proportions of boreal forest and tundra (mostly shrubs, together with spruce, fir and larch trees). At that time, the expansion of the SFM was clearly decreasing in intensity, and the recovery of the SFM’s current area was beginning, with visible population growth. The Holocene expansion was likely related to the recent expansion from the steppe habitats situated near the Russia–China border into the west, along rivers and human-modified habitats [15]. The spatial expansion model we fitted for SFM, with the demographic maximum during the glacial period, has already been applied to some other Eurasian or European species associated with steppe or forest-steppe biomes [112].

Researchers still do not agree on when the SFM first appeared in Europe. Most studies, usually based on the dating of scarce fossilized remains, emphasize that a crucial factor in the species’ expansion into Europe was the post-Neolithic deforestation during the Holocene [73,74,113,114]. Genetic studies in western Europe, however, suggest that the expansion of the SFM to Europe during the Neolithic was not natural, but induced by humans [30]. Nevertheless, some upper Pleistocene fossil records from south-eastern France, dated to about 17 ka [65], and a recently discovered record from north-western Bulgaria, dated to over 50 ka [71], indicate that the species may have expanded into Europe earlier than previously thought, or that this expansion was actually multiple colonizations. Knitlová and Horáček [6] initially assumed (based on fossil records) that the SFM colonized the western Palearctic (central and western Europe) during the middle Weichselian (MIS 4–3). Early Holocene records of this species in the Pannonian region of Slovakia [7], Belarus [21] and Italy [69] have all been dated to the Preboreal and Boreal periods (Greenlandian). This supports the hypothesis of a pre-Holocene expansion or multiple Pleistocene and early Holocene expansions, indicated by late Weichselian (MIS 2) records in Europe [65,71]. 

Our reconstruction of the historical distribution of the SFM has confirmed that potential refugia might have existed in south-western and southern Europe (that is, in the Balkans and the northern part of the Italian peninsula) (Figure 7). Most of these regions have been recorded as places with fossil records from the upper Pleistocene and early Holocene [7,23,31,32,73,74] (Appendix A). Nevertheless, there are no fossil records confirming the presence of glacial refugia in Caucassus during LGM. These discrepancies between the historical (the early Holocene) and current distributions of the species led to the formulation of the hypothesis that after a decline in SFM distribution during the LGM, the species expanded again, probably reverting to its previous, more extensive area of occupancy [6]. The BSP demographic history and MMD expansion models obtained in this study both support the hypothesis that the European population of the SFM expanded to Europe for the first time during the middle Weichselian (MIS 4–3), and then for a second time after the post-Neolithic deforestation after the bottleneck (Figure 5a, Appendix A). This past demographic reconstruction, with the peak of expansion falling in the middle and late Glacial (MIS 4–2) and the recent late Holocene expansion followed by the early/middle Holocene (postglacial) bottleneck, is supported by ABC analysis as one of the most probable scenarios (Figure 6a; Appendix A).

This deforestation, a consequence of climate changes during the LGM, and the resulting rapid reduction of the species’ range, probably explains the decline in the population growth rate during MIS 2, reaching the early and middle Holocene. The observed bottleneck of the Eurasiatic population C1, falling on late Glacial/early Holocene (~12–38 ka), is similar to the estimated time of disjunction between European and east Asiatic populations, dated on the early Holocene (<12 ka) or pre-LGM period (~39 ka), depending on the used molecular clock calibration [28,29]. The reconstructed vegetation pattern in Europe during the past 20,000 years indicates cyclical environmental changes during this period, a result of postglacial oscillations in climate. The last glacial vegetation in Europe was dominated by herbaceous plants (with *Poaceae* and *Artemisia* as the most common species), scrublands (with *Juniperus* and *Betula*) and boreal woodlands (with *Pinus* and *Betula*) [115,116]. During the Boreal and Atlantic stages, forest ecosystems replaced steppe and other non-forest biomes; dense deciduous forests dominated north-western and central Europe, and coniferous forests were dominant in southern regions of Europe and the Balkans [117].

During the late stages of the Holocene (mostly the late Holocene), the natural environment in Europe started to change, an effect of increasing human activity, which significantly affected biotopes [113,114,118]. This resulted mainly in deforestation, a favorable change for the SFM. SFM is a euryphagous species, but on an annual average, plant food represents about 70%, with a substantial part consisting of seeds of agricultural plants and weeds of the cultivated steppe [119]. This highlights the SFM’s habitat preference for deforested land. Indeed, the postglacial demographic trends of the SFM in Europe, as reconstructed in this study, correlate with postglacial changes in the distributions of herbaceous and grass taxa (Appendix A). This phenomenon was represented by two particularly interesting processes. First, during the early Holocene, the SFM’s range in Europe shrank, a likely result of the aforementioned changes in vegetation during this period; dense broadleaved and coniferous forests, disliked and avoided by the species, replaced its preferred steppe and scrubland biomes. Secondly, cyclical climatic changes and human activities during the Holocene affected vegetation, leading to significant deforestation and thus offering the SFM favorable conditions, which it took full advantage of during its most recent expansion in Europe during the Epiatlantic stage and the late Holocene. 

During glaciation, semi-open and open biomes dominated in northern Eurasia, thereby contributing to rapid expansions of other steppe species, which increased their western ranges during this period. Examples include *Mustela eversmanii* [120], *Spermophilus citellus* [121], *Cricetus* [122], *Sicista subtilis* [123] and *Saiga tatarica* [124], all of which had wider ranges during the upper Pleistocene than they have now. 

The methods implemented in this work can help to understand the reasons for recent population declines and to better reconstruct the history and demography of other widely distributed species, with still poorly recognized history, especially in the context of past climatic and environmental changes, that could shape their current genetic and population structure and initiated large-size expansion.

## 5. Conclusions

The reconstruction of the SFM’s historical demographic trends show that its history, in terms of its population and overall Eurasian range, was closely related to environmental and climatic fluctuations during the upper Pleistocene and early Holocene. The analysis of the SFM demographic history showed two populations that recently expanded: one from south-eastern Asia and the far east of Russia to the Korean Peninsula, and the other that expanded across the northern hemisphere as far as Europe. In the case of the south-east Asian population, the reconstructed demographic trends were correlated with sea level changes and the formation of favorable biomes on new land bridges formed by the exposed sea shelf. The westward expansion of the northern population, from north-eastern Asia through Siberia to central Europe, began after the last interglacial period. Having occurred during a period of relatively stable and warm climate in the second phase of the last Glacial period, the population growth of the SFM could be associated with the favorable environmental and climatic conditions at that time. In both cases, our results support the role of climate-induced environmental changes during the upper Pleistocene and early Holocene, as an important factor shaping history of the species. These results can help to understand the factors that influenced Pleistocene expansion and Holocene population declines of some other mammals, such as the European hamster, European ground squirrel, Steppe polecat or Saiga antelope, which had wider ranges during the upper Pleistocene than they have now.

The second important issue in this work is the recognition of the impact of man-made large-scale environmental changes during the Holocene on the species demographic history. In the case of the studied rodent, the Holocene expansion in Europe is considered to be the species’ second expansion after the population decrease during the Pleistocene–Holocene transition, affected by postglacial changes in northern environments. In the case of SFM, this recent expansion can be associated with the middle and late glacial human-induced changes in European biota. 

The research techniques used in this work, combining the analysis of genetic material with modelling of predictive past species distribution and a review of habitat changes over the upper Pleistocene and early Holocene constitute an effective method of reconstructing the history of SFM in relation to past climatic conditions, and identify the environmental factors that influenced past demographic trends and could have initiated large-scale or regional expansions, or bottleneck events, within site-specific populations.

## Figures and Tables

**Figure 1 genes-12-00642-f001:**
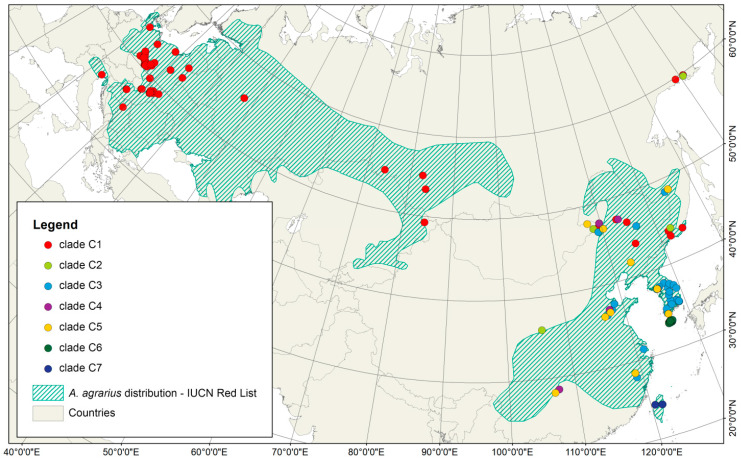
Origins of the samples used in the genetic analyses and their assignments to genetic clades of *A. agrarius* with geographic range of species, downloaded from IUCN Red List of Threatened Species. Version 2020-3.

**Figure 2 genes-12-00642-f002:**
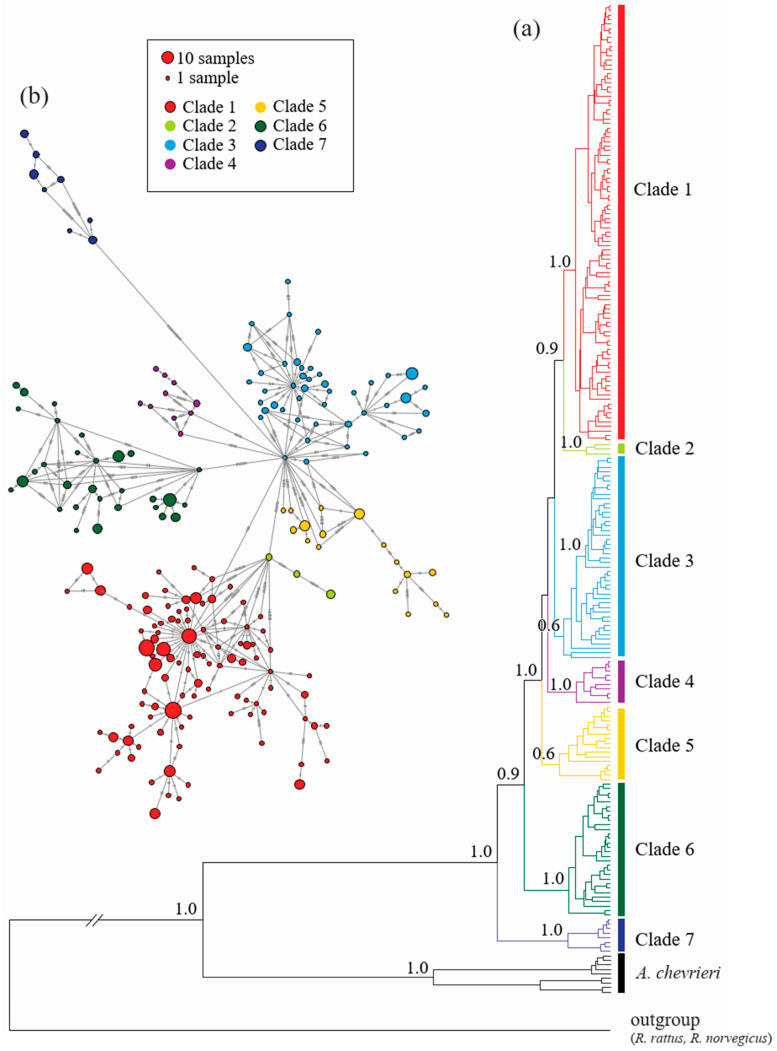
The minimum spanning network of *A. agrarius* generated in PopArt 1.7 (**a**) and phylogenetic tree of *A. agrarius* and *A. chevrieri* based on Bayesian inference (BI) in BEAST (**b**). Different colors represent separate genetic clades: red—clade C1, light green—clade C2, blue—clade C3, violet—clade C4, yellow—clade C5, dark green—clade C6, navy blue—clade 7.

**Figure 3 genes-12-00642-f003:**
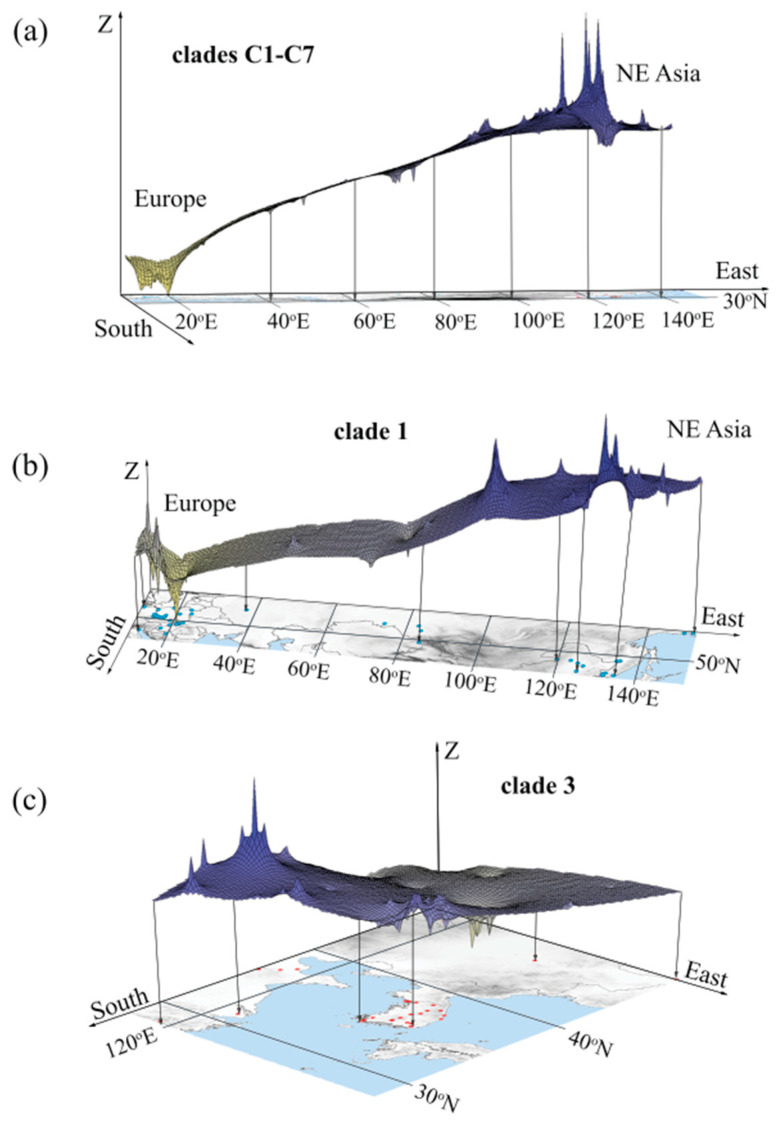
Spatial genetic divergence of *A. agrarius* in Eurasia (**a**) and within clades C1 (**b**) and C3 (**c**), visualized using the Landscape Shape Interpolation tool in Allele In Space, with three-dimensional surface plots. Surface plot heights (Z) reflect genetic distances between the corresponding populations.

**Figure 4 genes-12-00642-f004:**
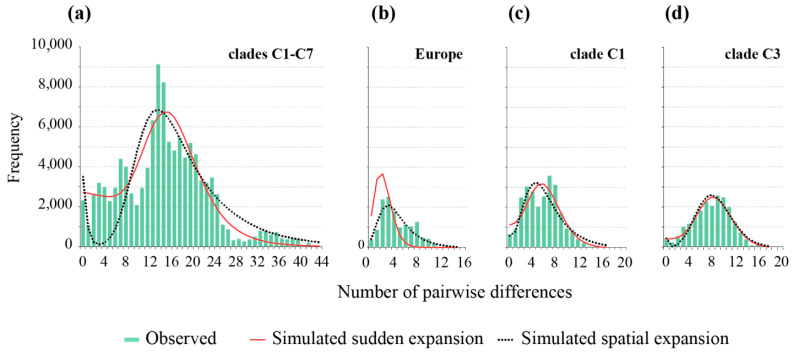
Reconstructed mismatch distributions (MMD) of *A. agrarius* in Eurasia (**a**), European clade C1 (**b**), clade C1 (**c**) and clade C3 (**d**), with the observed (histograms) and simulated distributions for the spatial (dotted black lines) and sudden (solid red lines) expansion models.

**Figure 5 genes-12-00642-f005:**
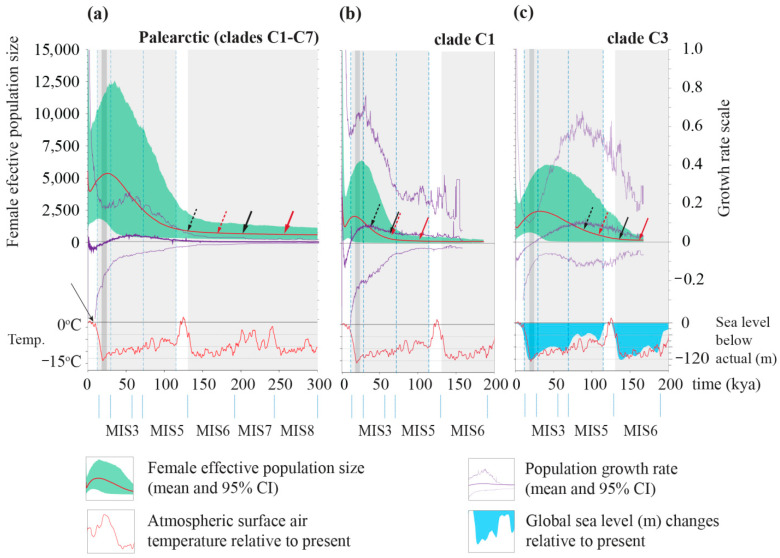
The reconstructed demographic history of *A. agrarius* obtained from the Bayesian skyline plot (BSP) in Eurasia (**a**), clade 1 (**b**), and clade 3 (**c**). Solid red lines—median, green areas—95% HPD confidence limits, violet lines—mean growth rates (with 95% HPD). The arrows indicate the estimated start of expansion based on the mismatch distribution estimation; black arrow—spatial expansion model, red arrow—pure demographic expansion model with either the standard evolutionary rate of 2.4% per Ma (solid line) or the fast evolutionary rate 3.6% per Ma (dotted line). The scales of the time axes (X) are the same. Vertical dashed blue lines indicate distinct last Glacial periods mentioned in text (late Weichselian—MIS 2, middle Weichselian MIS 3–4, early Weichselian MIS 5a–d). Grey area defines the glacial periods, with LGM marked in dark grey. The plot of temperature and sea level changes is based on [61] and the extent of glacial periods and Marine Isotope stages based on [62,63,64].

**Figure 6 genes-12-00642-f006:**
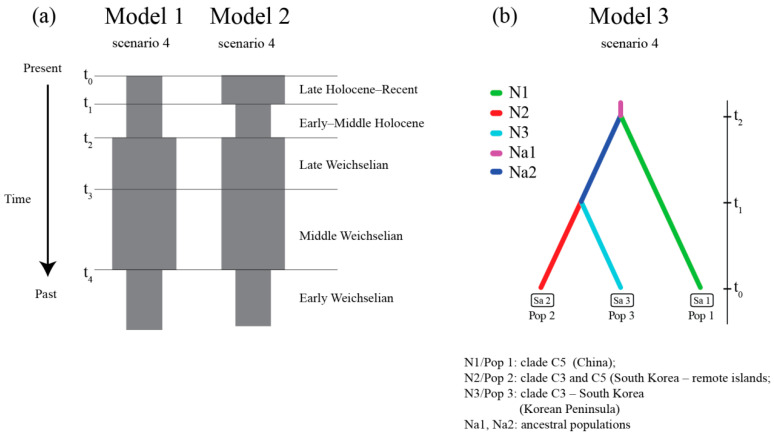
Representation of the most probable demographic scenarios for clades C1 and C3 (Model 1 and 2) (**a**) and evolutionary scenario of divergence the Korean population of *A. agrarius* (Model 3) (**b**) analyzed with the ABC method implemented in DIYABC.

**Figure 7 genes-12-00642-f007:**
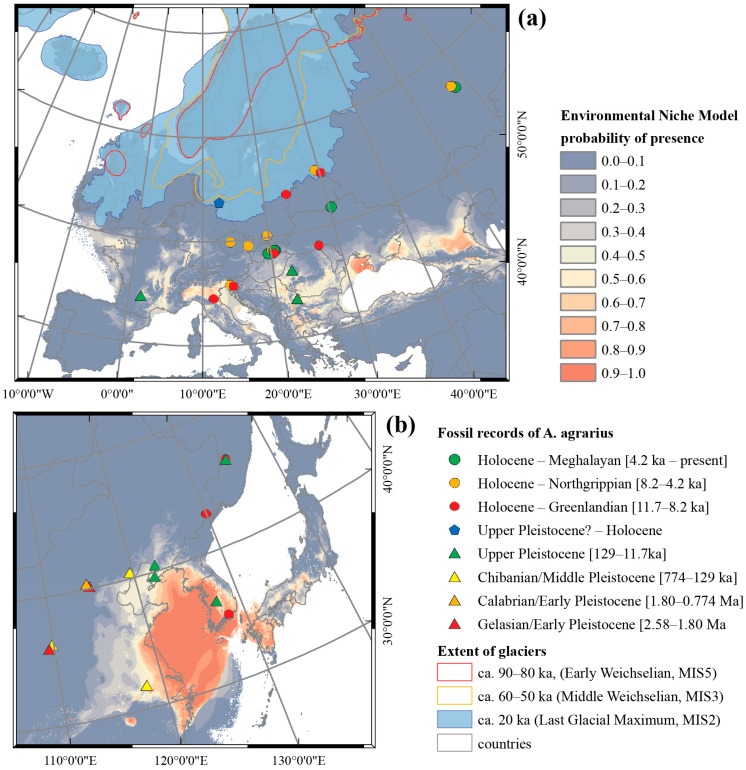
The environmental niche models (ENMs) for *A. agrarius* during the LGM in clades C1 (**a**) and C3 (**b**). The ENMs were generated in MaxEnt. Fossil records in Eurasia were based on the literature [6,21,65,66,67,68,69,70,71,72].

**Table 1 genes-12-00642-t001:** The genetic diversity indices of mtDNA (cyt b) of the populations and identified phylogenetic clades; *n*—number of sequences, *S*—number of segregating sites, *Eta*—total number of mutations, *k*—average number of differences, *h*—number of haplotypes, *Hd*—haplotype diversity, *π*—nucleotide diversity.

Population	*n*	*S*	*Eta*	*k*	*h*	*Hd* ± S.D.	*π* ± S.D.
*A. agrarius—Eurasia*	473	255	275	15.17	210	0.980 ± 0.003	0.0133 ± 0.0004
*Eurasia—mainland*	392	209	227	11.46	172	0.972 ± 0.005	0.0101 ± 0.0002
Asia	307	230	243	17.80	143	0.960 ± 0.007	0.0156 ± 0.0004
*Asia—mainland*	226	179	191	13.35	105	0.929 ± 0.013	0.0117 ± 0.0002
Europe	166	78	81	4.52	67	0.973 ± 0.004	0.0040 ± 0.0002
Clades							
Clade 1	226	110	118	5.71	97	0.974 ± 0.004	0.00501 ± 0.00020
Clade 2	59	12	12	1.11	3	0.133 ± 0.060	0.00099 ± 0.00044
Clade 3	65	95	96	7.86	44	0.979 ± 0.009	0.00690 ± 0.00035
Clade 4	15	25	26	6.54	10	0.895 ± 0.070	0.00574 ± 0.00089
Clade 5	28	47	47	9.36	17	0.947 ± 0.025	0.00821 ± 0.00049
Clade 6	64	63	63	7.30	30	0.958 ± 0.011	0.00640 ± 0.00031
Clade 7	17	21	21	8.12	8	0.897 ± 0.042	0.00712 ± 0.00108

**Table 2 genes-12-00642-t002:** Analysis of molecular variance (AMOVA) for *A. agrarius* using mtDNA between (1) seven clades C1–C7, (2) three groups (Eurasia, Taiwan and Jeju Island) from these clades, and (3) three groups (Eurasia, Taiwan and Jeju Island) stratified according to countries/populations. *—statistically significant values (*p* < 0.05).

Source of Variation		d.f.	Sum of Squares	Variance Components	Percentage of Variation	Fixation Indices
(1)	Among seven clades	6	2196.1	6.433 * Va	68.3	FST: 0.683 *
	Within clades	467	1393.0	2.983 * Vb	31.9	
(2)	Among three groups	2	1045.5	4.157 * Va	30.0	FSC: 0.613 *
	Among five clades within Eurasia (excluding islands)	4	1150.5	4.727 * Vb	39.8	FST: 0.749 *
	Within clades	467	1393.0	2.983 * Vc	25.1	FCT: 0.350 *
(3)	Among three groups	2	1045.5	6.114 * Va	50.1	FSC: 0.377 *
	Among nine countries within Eurasia (excluding islands)	8	790.0	2.292 * Vb	18.8	FST: 0.689 *
	Within population	467	1753.5	3.787 * Vc	31.1	FCT: 0.501 *

**Table 3 genes-12-00642-t003:** The time of the most recent expansion, estimated using Arlequin 3.5 separately for the populations and clades of *A. agrarius*, based on the sudden demographic and spatial expansion models for three different mutation rates with 95% CI values. Estimated evolutionary rate: (^1^) 0.024/Ma [48], and (^2^) 0.027–0.036/Ma [49,50].

Groups	Sudden Expansion Model				Spatial Expansion Model			
	Tau Est. Val. (95% CI)	Evolutionary Rate (per Site per Million Years)			Tau Est. Val. (95% CI)	Evolutionary Rate (per Site per Million Years)		
		2.4 × 10^−2 1^	2.7 × 10^−2 2^	3.6 × 10^−2 2^		2.4 × 10^−2 1^	2.7 × 10^−2 2^	3.6 × 10^−2 2^
		Expansion Time (ka) Mean (95% CI)				Expansion Time (ka) Mean (95% CI)		
Populations								
**Eurasia**	14.1 (8.5–29.9)	257.7 (156–546)	229.0 (138.7–485.4)	171.8 (104–364)	11.0 (7.5–26.0)	201.5 (137.9–474.7)	179.1 (122.6–422.0)	134.3 (91.9–163.3)
**Asia**	-	-	-	-	13.9 (10.1–21.7)	154.4 (185.0–396.2)	226.1 (164.4352.2)	169.6 (123.3–264.1)
**Asia Mainland**	-	-	-	-	15.5 (10.9–18.2)	282.9 (199.1–333.2)	251.4 (177–296.2)	188.6 (132.7–222.2)
**Europe**	-	-	-	-	2.1 (0.9–5.7)	38.9 (16.4–103.3)	34.6 (14.6–91.9)	25.9 (10.9–68.9)
**Clades**								
**Clade 1**	5.4 (2.8–11.1)	99.5 (51.7–202)	88.4 (46.0–179.5)	66.3 (34.5–134.7)	3.5 (1.7–7.3)	63.7 (30.7–134.3)	56.6 (27.3–117.4)	42.4 (20.4–89.5)
**Clade 3**	8.9 (5.3–11.5)	162.6 (96–209.4)	144.5 (85.3–186.1)	108.4 (64.0–139.6)	7.4 (5.2–10.3)	134.4 (95.1–187.6)	119.5 (84.6–166.7)	89.6 (63.4–125.0)

## Data Availability

All complete cytb (1140 bp) haplotypes of newly sequenced specimens of *A. agrarius* used in analyses have been deposited in Genbank database (GenBank accession number MT113485-MT113569).

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
