# Peer review of "Late Pleistocene Expansion of Small Murid Rodents across the Palearctic in Relation to the Past Environmental Changes"

_genes, 2021, doi:10.3390/genes12050642_

Round 1

Reviewer 1 Report

This study examined the demographic change and phylogenetic (genetic) structure of the striped field mouse Apodemus agrarius across its distribution from Europe to East Asia based on the mitochondrial cytochrome b gene sequences. The authors determined this locus for 184 individuals from central Europe (153) and eastern Mongolia (31) and examined the above-mentioned population and evolutionary characteristics together with the previously reported sequences. This study clarified that two islands (Jeju and Taiwan) had each distinct haplotype, mainland Asian populations had specific four lineages that are not observed in Europe, and European populations had a widely distributed lineage that is observed across Eurasia (Europe and east Asia). The demographic analyses based on mismatch analysis, Bayesian Skyline Plot analysis, and some indices of genetic diversity such as Tajima’s D, Fu’s Fs, and Ramos-Osins and Rozas R2 statistic showed that two major clades C1 and C3 experienced demographic expansions from east Asia to Europe and Korea, respectively. Lower genetic diversities in populations in Europe and Korea also support that these areas were frontiers of the population expansions. Furthermore, using Species Distribution Modeling (SDM) analysis, they inferred the suitable habitat of A. agrarius during the last glacial maximum and clarified that northeastern China, a part of Japan (not mentioned), and southern parts of Europe possessed possible refugia during this harsh period. Based on the evidence, the authors discussed the relationships between the demographic expansion and environmental changes in the Late Pleistocene.

I highly evaluate that their molecular evolutionary, demographic, and SDM analyses are performed well, meeting the standard of the up-to-date analyses of this kind of research field. It would be also good that recently discussed time dependency of evolutionary rate was also considered for their time calibrations. The results they obtained are interesting although using only one marker may not be enough in the era of the genomic analyses. I suppose that this manuscript could be publishable if the authors consider my concerns below. I would like to see further improvement of this manuscript before recommending publication. Below I listed my major and minor concerns.

Major concerns

  1. Introduction is not well oriented toward the goal of this study that aims to clarify the past demographic change and genetic structure of A. agrarius. The authors should concentrate on the explanation of the background that is relevant to these issues and not mention too much about adaptation that was not examined in this study. Thus, the introduction section should be restructured.
  2. In SDM analysis, the Japanese archipelagos were predicted to be potentially suitable habitat for A. agrarius as shown in Fig. 7. However, this species is not present in the archipelagos now (except for the Senkaku Island). It could be attractive for the Japanese researchers if you explain the reason why this species is not currently present. I suppose that the timing of the geological division between the continent and Japan (Korea/Tsushima straits) or the existence of the two congeneric species that may prevent A. agrarius from colonizing the archipelagos could explain the absence. The literature below may be useful.

Sato JJ (2016) A review on the process of mammalian faunal assembly in Japan – insight from the molecular phylogenetics. In (M. Motokawa and H. Kajihara, eds) Species Diversity of Animals in Japan. Springer Japan pp. 49-116.

  1. The evolutionary and demographic history of this species should be compared with other similarly distributed organisms. The authors discussed only a little (L561-564, L626-628), but it would be important to understand if the obtained trends are specific to this species or more universal.

Minor concerns

L38-39: I don’t think that “constant change” is appropriate expression. Glacial periods have come cyclically. Some environmental change would be abrupt. It is difficult for us to assume that the environmental change is constant.

L47: Adaptation to environmental changes via glacial and non-glacial cycles and via agricultural and other human activity would be different in time scale. The authors should not combine them under the same context. In other word, even if we clarify the adaptation of the mice to human-altered landscape, we could not understand the ability of the mice adaptation to larger scale changes like glacial cycles. The opposite is also true.

L48-62: I can point out the same thing here. Adaptation of the invasive species does not fit to the time scale that the authors examined in this study. The authors explained adaptation here, but they did not examine the adaptation in this study. They inferred the past demographic change and genetic structure of the wood mice based on the possibly neutral genetic markers. The authors should tone down regarding adaptation.

L128: I am not sure if the birth-death process is appropriate tree prior in examining intraspecific diversity because this tree prior is for the analysis of interspecific relationships.

L131-132: The authors explained that they partitioned the cytb data into 1st, 2nd, and 3rd codon positions. However, only one model was described. Did they mean that they used only one substitution model with different parameters for each codon position? Didn’t they use multiple substitution models for each codon position?

L135: Better to note indices for the convergence (ASDSF and PSRF).

L138: mtDNA sequences > cytb sequences?

L140: countries > populations? Genetic diversity of the country is not scientific.

L242-244: How was the results for the remaining samples (361 samples – 184 sequence-detemined individuals)?

L281: Table 1 should be more understandable in terms that European and Korean populations have lower genetic diversity. It would be an important supportive evidence for the direction of the population expansion.

L346-355: Also, for C3, the time for the expansion should be described. It would especially be necessary to understand the origin of the Korean lineages as a result of the population expansion.

L367-384: In this paragraph, the timing of the expansion of ancestral populations are described. The timing is earlier for C1 than C3, which is different estimates from those described above. Are these explanations here correct?

L400-401: Fig. 7b is explained earlier than Fig. 7a. Is it OK?

L413: “very long periods of isolation” is a vague expression. Islands nearby the continent like Jeju or Taiwan should not have been isolated for a long time. These islands should have been connected to the mainland several times. There should therefore have been multiple colonization events into these islands as indicated in Hosoda et al. (2011). Probably, both ancient and young lineages exist in these islands. It is also the case in Japan (Sato 2016; see above for literature information).

Hosoda T, Sato JJ, Lin L-K, Chen Y-J, Harada M, Suzuki H (2011) Phylogenetic history of mustelid fauna in Taiwan inferred from mitochondrial genetic loci. Canadian Journal of Zoology 89: 559–569.

L429: I cannot understand why “promiscuity” can explain the weak isolation by distance. Such a weak IBD may be related to some barriers across the distribution. Are there any potential barriers that may hinder the gene flow and weaken the IBD between Europe and Siberia? Anyway, L427-431 is not relevant to the main point in this paragraph (IBD at larger scale) and can be omitted.

L437-439: What correlations? Please write more concretely.

L444: How different? It would be better to highlight this difference in Results section.

Reviewer 2 Report

I found the study interesting and important. Most of my comments are for the interpretation of your results and minor corrections needed.

Age models and terminology: Since Weichselian is the regional European glacial unit, please define it in terms of absolute time and MIS stages widely used in paleogeography.

Since you conducted a phylogenetic analysis and discuss times of diversity changes, it would be logical to calibrate your phylogenetic tree and estimate times of divergences between and within clades using molecular clocks, estimate times for TMRCA of different clades, etc. 

Please discuss data of Latinne et al. (2020) on the divergence times between western and far eastern populations in the light of your data. It would also be a good idea to discuss  possible contaminations of the historical clade structure by 20 century invasions when a large ammounts of agricultural traffic travelled along the trans-Siberian railway in western and eastern directions. Note the example of the isolated Magadan record of SFM originated by human-aided invasion.

Note that east-west expansion and expansion within Europe are two different stories. The former was only possible during interglacial time (Eemian, MIS5e). The latter (only in southern Europe) could occur during  glacial time. The problem of your study that the most promising refugial areas in south Europe (Bulgaria, Romania) are insufficiently studied.

Your environmental niche modelling obviously suffers from insufficient attention to biotopic and climatic constraints of SFM: during LGM many areas marked in you European maps as refugial were likely not suitable for SFM because of high aridity and low temperatures.

Figure 6: For Fig.6a indicate the clades C1 and C3 in the figure and figurу caption.

Figure 7: adjust the boundaty ages of the Quaternary stratons to the latest version of International Strtigraphic Chart (stratigraphy.org)

S5 Fig. 1. Indicate the reference to the information source on glacial expansion limits.

Round 2

Reviewer 1 Report

The manuscript has been much improved. Only I can suggest are like below.

L73: “Micromammalia” is not common. “Small mammals” is enough.

L85: SFM is also observed above. Define it at the place for the first mention.

L118: > climate-induced?

L382-383: I cannot confirm that C1 showed an earlier growth than C3 in Fig. 5.

L432: kKorean > Korean

L434-435: Korea is not in SE Asia, but E Asia.

L472: Again, “very long” is not scientific. At least “very” should be removed.

L648: > At that time ??

L680: formulate > formulation of [English should be checked throughout the manuscript]

Fig. 1: Probably it is easier to read from C1 to C7 from top to bottom in the legend. A. agrarius should be italic.

Table 2: A. agrarius should be italic in the legend.
